# On the Safety Concerns of Deploying LLMs/VLMs in Robotics Highlighting the Risks and Vulnerabilities

## Abstract

*In this paper, we highlight the critical issues of robustness and safety associated with integrating large language models (LLMs) and vision-language models (VLMs) into robotics applications. Recent works have focused on using LLMs and VLMs to improve the performance of robotics tasks, such as manipulation, navigation, etc. However, such integration can introduce significant vulnerabilities, in terms of their susceptibility to adversarial attacks due to the language models, potentially leading to catastrophic consequences. By examining recent works at the interface of LLMs/VLMs and robotics, we show that it is easy to manipulate or misguide the robot's actions, leading to safety hazards. We define and provide examples of several plausible adversarial attacks, and conduct experiments on three prominent robot frameworks integrated with a language model, including KnowNo [40], VIMA [21], and Instruct2Act [20], to assess their susceptibility to these attacks. Our empirical findings reveal a striking vulnerability of LLM/VLM-robot integrated systems: simple adversarial attacks can significantly undermine the effectiveness of LLM/VLM-robot integrated systems. Specifically, our data demonstrate an average performance deterioration of 21.2% under prompt attacks and a more alarming 30.2% under perception attacks. These results underscore the critical need for robust countermeasures to ensure the safe and reliable deployment of the advanced LLM/VLM-based robotic systems.*

## 1. Introduction

The advent of large language models (LLMs) and vision-language models (VLMs) has enabled robots to perform various complex tasks by enhancing their capabilities for natural language processing and visual recognition. This can increase their benefits for different applications, including healthcare [17, 27, 36], manufacturing [48, 50], and service industries [3, 11]. However, incorporating LLMs/VLMs into a robotic system can introduce unprecedented risks, primarily

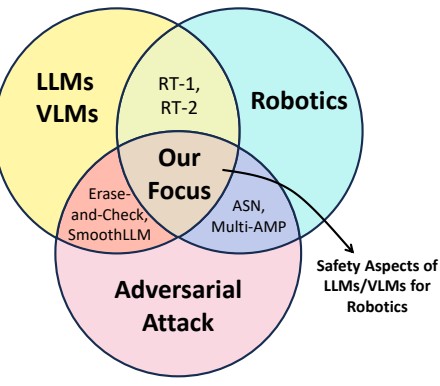

Figure 1. Our experiments expose vulnerabilities in state-of-the-art LLMs/VLMs for robotics, particularly to adversarial attacks, underscoring the need for further research to ensure the safety and reliability of using language models in robotic applications.

due to inadequate defense mechanisms. For instance, the hallucination and illusion of language models [14] could affect a reliable understanding of the scene, leading to undesired actions in the robotic system. Another source of risk comes from the failure of LLMs/VLMs to address the ambiguity of contextual information provided by text or images [35, 52]. Since the current language models usually follow a template-based prompt format to execute a task [16, 29], the lack of flexibility in addressing the variants and synonyms of natural languages could also contribute to the misunderstanding of prompts [24, 43]. Moreover, using multi-modality in prompt input increases the difficulty of context understanding and reasoning, which could lead to a higher failure risk [8, 18]. In practical applications, those risks would pose significant challenges to the robustness and safety of robotic systems.

Our goal is to analyze the trustworthiness and reliability of language models and robotics. In that regard, we aim to increase awareness regarding the safety concerns of the state-of-the-art language models for robotics applications via extensive experiments. We show that further research is needed on this topic to safely deploy LLM/VLM-based robots for real-world applications. Our primary focus is to

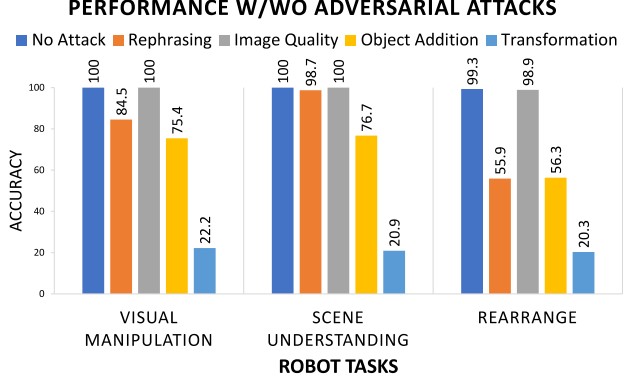

Figure 2. **Showcases of Successful Attacks to LLMs/VLMs in Robotic Applications.** The manipulator could successfully execute the pick-and-place (*Visual Manipulation*) task given the original prompt. However, when applying adversarial attacks, like the prompt rephrasing attack on adjectives, the information conveyed by rephrased prompts may be misunderstood by the robot system and lead to an incorrect operation, *e.g.* pick up the incorrect object and place it to an incorrect location.

Figure 3. To provide a preview of our findings, we showcase the reduction in accuracy of the LLMs/VLMs used in robotics, under various adversarial attacks. These results are presented across three different tasks: *Visual Manipulation* (pick and place), *Scene Understanding* (move objects with specific textures to target place given the scene image), and *Rearrange* (move objects to target places given the scene image), with the accuracy decrements averaged for each category of attack. Task details can be found in Section 8 in the Supplementary Material.

provide evidence of how the inherent complexities and learning mechanisms of LLMs/VLMs in robotics can improve or hurt the performance: while they introduce sophisticated functionalities, they also expose these systems to new vulnerabilities [12, 14, 31]. Adversarial attacks can lead to unexpected and potentially dangerous outcomes, particularly in scenarios where robotic decisions and actions have critical safety implications.

**Main Results**: In this paper, we conduct an extensive analysis of current applications and potential attack vectors and emphasize the critical need for robust security frameworks and ethical guidelines. We show that ensuring the safe deployment of LLM/VLM-enhanced robotics is not only a technical challenge but also a moral imperative, requiring concerted efforts from researchers, practitioners, and policymakers. Our main contributions include:

**1. Highlighting the vulnerabilities and safety concerns of using LLMs/VLMs in robotics.** We conduct an extensive literature review of recent LLMs/VLMs integrated robotics systems and provide an in-depth analysis of their vulnerability to adversarial attacks. To the best of our knowledge, ours is the first work to specifically address and discuss vulnerabilities in an LLM/VLM-based robot system.

**2. Design of adversarial attacks on LLM/VLM-based robotics systems.** We define and categorize adversarial attacks on LLM/VLM-robot integrated systems, classifying them into prompt and perception attacks based on our analysis. For each attack category, we outline various potential attack methods, along with detailed definitions and illustrative examples.

**3. Empirical analysis.** We apply and assess the adversarial attacks, across all the categories, on three state-of-the-art LLM/VLM-robot approaches, including KnowNo [40], VIMA [21], and Instruct2Act [20]. We propose several evaluation experiments for each attack and show that our adversarial attacks deteriorate the success rate of the LLM/VLM-robot integrated system by 21.2% under prompt attack and 30.2% under perception attack on average for manipulation tasks.

**4. Highlighting key open questions.** We highlight some key issues that need to be addressed by the research community to ensure the safe, robust, and reliable integration of language models in robotics based on the insights and findings of our study.

## 2. Literature Review

### 2.1. Language Models for Robotics

**Manipulation and Navigation Tasks.** The integration of Large Language Models (LLMs) and Vision Language Models (VLMs) with robotics marks a significant advancement in embodied AI [9, 10, 15]. This fusion allows robots to leverage the commonsense and inferential capabilities of

language models in decision-making tasks. According to the criteria outlined in recent research [25, 41], the application of these models in robotics primarily encompasses navigation and manipulation tasks. Navigation tasks involve using Vision-Language Models (VLMs) trained on extensive image datasets, enabling robots to understand human instructions, recognize objects and their positions, and navigate effectively. These capabilities also aid in detecting out-of-domain objects and pinpointing targets within their spatial perception [19, 34, 38]. In contrast, manipulation tasks [4, 5, 21, 32, 45] involve processing human language instructions and using visual perception to locate objects within a scene. Here, large multi-modal models combine visual and language inputs to generate actions for robotic manipulators, aiding in scene understanding, grasping, and object arrangement in simulated and real-world environments. **Reasoning and Planning Tasks.** Another key classification criterion is the complexity of tasks undertaken by large models, which span from basic perception to advanced reasoning and planning. In perception-based tasks, these models either autonomously gather training data through scene observation without human labeling [51], or learn about unseen objects from expansive Internet-sourced datasets [46]. Conversely, in reasoning and planning tasks, the models engage in sophisticated decision-making, drawing on their scene comprehension and inherent commonsense knowledge [4, 30, 37]. Research efforts have enhanced these models' capabilities, such as pre-training for task prioritization [1] and converting complex instructions into detailed tasks with rewards [53]. These models facilitate human-in-the-loop decision-making, where human input refines robot demonstrations. Innovative frameworks have been developed that enable robots to comprehend and learn from human demonstrations and instructions [44], further integrating large multi-modal models in task understanding. Additionally, [40] proposed a framework that allows robots to seek additional guidance from human overseers when faced with decision-making uncertainties. Despite the extensive research and development in LLM/VLM-robot integration, there has been a notable lack of attention to the potential risks, especially the threat of adversarial attacks on advanced robotic systems. This oversight could lead to severe consequences if exploited by malicious actors.

## 2.2. Adversarial Attacks on Language Models

Adversarial attacks are inputs that reliably trigger erroneous outputs from language models [47]. These attacks encompass diverse strategies such as Token Manipulation, Gradient-based Attack, Jailbreak Prompting, and Model Red-Teaming. Token Manipulation, for instance, involves altering model predictions through synonym replacement, random insertion, or swapping of the most influential words [22, 28, 33]. Gradient-based attacks exploit the model's own gradients to find vulnerabilities. Jailbreak Prompting, a more sophisticated technique, involves crafting prompts that bypass model restrictions, while Model Red-Teaming tests model robustness against various adversarial inputs. Studies by [23, 55] have delved into the creation of universal adversarial triggering tokens, examining their efficacy as suffixes added to input requests for language models. [13] research highlights the exploitation of language models to analyze external information, such as websites or documents, and introduces adversarial prompts through this channel. [12, 14, 31] revealed vulnerabilities in language models by demonstrating the limitations of one-dimensional alignment strategies, especially when dealing with multi-modal inputs.

## 2.3. Safety Concerns of LLMs/VLMs in Robotics

Substantial evidence in current literature underscores the effectiveness of LLMs/VLMs in robotics, highlighting their superior performance in various applications [49, 54]. For instance, these models support robots with enhanced reasoning capabilities, enabling them to act effectively in real-world scenarios. Furthermore, they empower robotic systems with the ability to process and understand natural language instructions, a crucial aspect of human-robot interaction [2]. Despite these advancements, our review of the literature reveals a notable gap: to the best of our knowledge, there is a lack of comprehensive studies addressing the potential vulnerabilities and risks associated with the deployment of language models in robotics. Our work aims to fill this gap by being the first to rigorously focus on this aspect, providing empirical evidence that highlights the risks and challenges of utilizing language models with robotics.

## 3. Highlighting the Risks: LLMs/VLMs for Robotics

In this section, we delve into the sophisticated architecture of a robotic system integrated with language models [20, 21]. The two key input modalities include: **Visual Inputs** (RGB images or segmentation) and **Textual Prompts** (human instructions). These high-level inputs are translated by the vision-language models (VLMs) into practical and actionable commands for the robot. This process enables the robot with a nuanced contextual understanding to intelligently interpret human instructions and visual cues. After receiving the commands, the robot interacts with the physical world, makes new observations, receives feedback from the surroundings, and then processes the information by VLMs again.

### 3.1. Vulnerabilities

In the system architecture outlined in Figure 4, the vision-language model plays a crucial role, bridging between complex environmental data, user instructions, and the robot's simpler, executable commands. Nevertheless, this critical

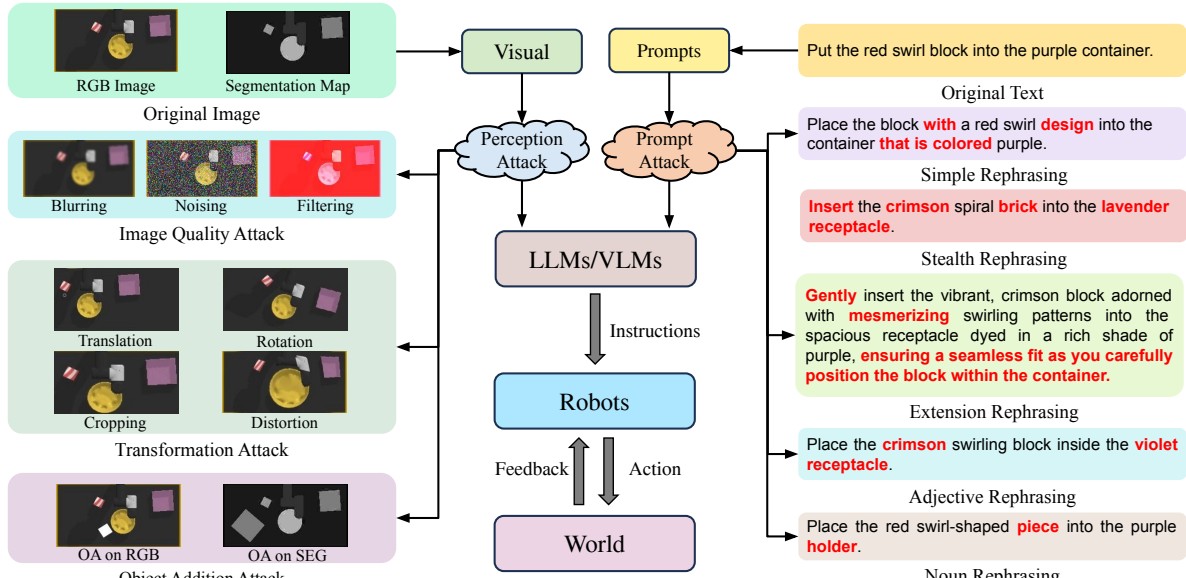

Figure 4. **Multi-modal Attacks to LLMs/VLMs in Robotic Applications.** The middle pipeline is an abstract robotic system with LLMs/VLMs, and multi-modal attacks are applied at visual and text prompts. The left-hand side provides different attacks to images, such as reducing image quality, applying transformation, and adding new objects. The right-hand side shows different types of attacks in text, including simple rephrasing, stealth rephrasing, extension rephrasing, and rephrasing of adjectives and nouns.

interpretative role exposes the model to potential vulnerabilities from adversarial attacks. These weaknesses include:

**Inaccurate Data Acquisition or Interpretation.** Failure of the model to gather or understand perceived data correctly.

**Misinterpretation of Human Instructions.** The potential for incorrectly interpreting human directives.

**Erroneous Command Generation.** The risk of formulating impractical or incorrect commands for the robot.

Within the spectrum of possible avenues for adversarial attacks, our attention is concentrated on two primary vulnerabilities. These vulnerabilities facilitate low-cost and easily implementable adversarial attacks, which could precipitate critical malfunctions in the entire robotic system. Such attacks can be achieved by simply modifying the inputs fed into the vision-language models, underscoring the need for heightened awareness and robust countermeasures. We discuss two types of them as follows:

**Prompt Input.** Most prompts provided to the vision-language models that are integrated with the robot system are highly template-based and depend on pre-defined keywords for semantic understanding [20, 21, 40]. Our analysis reveals that these prompts adhere to a formulaic pattern: $Action + BaseObject + TargetObject$. The placeholders for both $BaseObject$ and $TargetObject$ are constrained to a composition that includes an adjective describing the object's properties and a noun identifying the object, such as 'Put the red swirl block into the purple container', 'Put the green and purple stripe star into the yellow and purple polka dot pan'.

This composition is derived from a limited, pre-established vocabulary, exhibiting a notable deficiency in diversity.

**Visual Input.** The vision-language models primarily receive their visual inputs from the robot's sensory equipment, such as an RGB camera, but it may also process additional data like segmentation maps derived from the RGB images. For the robot system to perform accurately, the integrity and quality of this image data are crucial. They enable the robot to precisely localize objects and clearly understand its surroundings. However, the semantic interpretation of these images can be easily compromised. In Figure 4, simple manipulations such as image rotation or distortion can disrupt the logical connection between objects in the perceptual field, thereby posing a significant threat to the functionality of the vision-language models within the robotic system.

## 4. Methodology

Based on the vulnerabilities outlined in Section 3, we can categorize our proposed attack into three distinct approaches: *Prompt Attack*, *Perception Attack*, and *Mixture attack*. We discuss them in detail as follows.

### 4.1. Prompt Attack

The prompt attack is to rephrase the initial instruction prompt, with the aim of challenging the interpretative ability of the robot system. As highlighted in Section 3.1, the instruction prompts are predominantly formatted as $Action + BaseObject + TargetObject$. The prompt attacks aim

to either disorganize such structure by rearranging the components and introducing redundant words or directly attach prompt understanding by replacing the keywords, including the adjectives that describe object properties and the nouns corresponding to the object names, with their synonyms. We categorize the prompt attacks into the following five types as described in Figure. 4 and below:

**Simple Rephrasing** involves rephrasing the prompts into a different structure while preserving the original meaning.

**Stealth Rephrasing** entails delicately reshaping the underlying meaning of prompts while preserving their surface meaning through subtle rephrasing.

**Extension Rephrasing** involves elaborating the prompts using more words while preserving the original meaning.

**Adjective Rephrasing** involves replacing adjectives within the prompts that describe object properties, such as color, patterns, and shapes, while preserving the original meaning.

**Noun Rephrasing** involves replacing the nouns in the prompts, such as '*bowl*' and '*boxes*', while preserving the meaning of the objects.

Additionally, prefixes used for rephrasing the prompts in these attacks and their outcomes are detailed in Table 3 and 4 in Section 9 in the Supplementary Material.

### 4.2. Perception Attack

The perception attack applies modifications to the visual observation of the robotic system perceived from the environment, There are multiple perception attack approaches, categorized under 3 general perspectives. Examples of these attacks are presented in Figure. 4.

**Image Quality Attack** is to degrade the quality of the images that the robot system perceived, which includes: **(a) Blurring.** Implementing Gaussian blurring on the RGB images captured by the robot system. **(b) Noising.** Introducing Gaussian noises into RGB and segmentation images. **(c) Filtering.** Adjusting the pixel values in a specific RGB channel to their maximum.

**Transformation Attack** involves applying transformation onto images to change the properties of the objects within the robot's perceptual field. Attacks in this genre include: **(a) Translation.** Shifting the image along the $x$ and $y$ axes to change the position of objects in the view. **(b) Rotation.** Rotating the image around its center point and altering the orientation of objects within the scene. **(c) Cropping.** Cropping part of the image and resizing it to change the context or focus of the image. **(d) Distortion.** Applying a distortion matrix to the image that warps the appearance of objects in the scene, affecting their perceived shapes and positions.

**Object Addition Attack** involves inserting a fictitious object into the image perceived by the robot, an object that does not exist in the actual environment. Object addition attacks include: **(a) Object Addition in RGB.** Selecting a random rectangular area in the RGB image and fill it with white. This creates the illusion of an additional object within the scene. **(b) Object Addition in Segmentation.** Choosing a random rectangular area in the segmentation image and filling it with a random, pre-existing object ID. This introduces a new, artificial object into the segmentation map. Detailed information on the implementation of these perception attacks can be found in Table 5 in Section 10 in the Supplementary Material.

### 4.3. Mixture Attack

Considering the prompt and perception attacks we have outlined, adversaries targeting the robotic system could employ a combination of two or more such attack approaches to further degrade the system's performance. For instance, they might simultaneously rephrase the adjectives in the prompts and apply distortion to the images. In our experiments, we conduct a detailed analysis of the performance differences of the robot system under various combined attacks.

## 5. Experimental Evidence

### 5.1. Evaluation Plans and Metrics

Among all works at the intersection of language models used in robot systems, we choose the following three models, KnowNo [40], VIMA [21] and Instruct2Act [20], to evaluate our adversarial attack approaches, while all models are applied for object manipulation or arrangement tasks with robot manipulators and visual perception based on some visual reasoning abilities from language models. The details of the comparisons are discussed in Section 7 given in the Supplementary Material. We show some failure cases in Section 12 in Supplementary Material and GIF animations in the attachment.

**Evaluation Metrics.** The success rate given in percentages is the metric we use to evaluate and compare the difference in performance before and after adversarial attacks for each of the works we mentioned above. For KnowNo, we run 500 calibration examples before execution as the in-context learning for LLM. For VIMA and Instruct2Act that use VIMA-Bench, we evaluate both approaches under adversarial attacks over 3 tasks with 3 difficulty levels. We run each adversarial attack over each task for each model for 150 iterations allowing 5 possible attempts when executing tasks and computing the overall success rate throughout the whole evaluation procedure.

### 5.2. Results Analysis with Textual Prompt

We first perform attack experiments on KnowNo [40] using textual prompts as its input without any visual inputs. Only prompt attack is allowed in this scenario. Results are provided in Figure 5.

KnowNo is robust under **Simple and Extension Rephrasing** without much accuracy reduction. The rationale be-

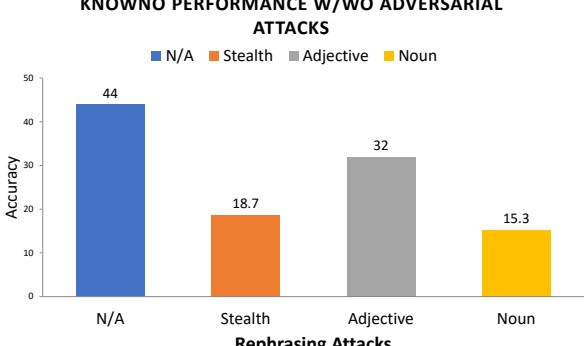

Figure 5. **Prompt Attack Results of KnowNo [40] over the pick-and-place manipulation task.** All prompt attack results are presented and compared with the no-attack baseline. **Remark.** The KnowNo framework is more vulnerable under stealth rephrasing attacks and noun rephrasing attacks.

hind this stems from the fact that both rephrases provide more explanations of the sentences, the information helps the language model to easier find more important information about the scene. **Stealth Rephrasing** reduces the accuracy to 18.7%, revealing its strong ability to confuse the LLM when understanding the prompts. **Adjective Rephrasing** reduces the accuracy to 32.0%, because different adjectives provide different properties of the objects. This operation confuses the model from understanding object texture and scene information correctly. **Noun Rephrasing** reduces the accuracy to 15.3% after attack. Similar to adjective rephrasing, noun rephrasing uses synonyms to change the description away from the real objects. Since the nouns are typically the nucleus of the compound referring to objects, the rephrasing attack targeting nouns is more effective than others. Thus, LLM cannot understand the scene correctly.

**Remark.** Overall, the prompt attacks targeted specific, essential components and the prompt structures that are decisive in context-understanding procedures, significantly deteriorating the performance of the robot language model, while attacking the nucleus component of the compound like nouns is more effective than others. This highlights the heavy reliance of current language models in robotics on identifying keywords from templates or training data for decision-making. Considering the inherent ambiguity of human language and workspace uncertainty in robot systems, such vulnerabilities, which are easily detectable and accessible, raise the potential for cost-effective adversarial attacks. Attackers only need to target adjectives and nouns describing objects in the scene or break the structure of the prompt by altering its meaning subtly, which can result in significant losses in real-world robot applications.

## 5.3. Results Analysis with Multi-modal Prompt

We perform both prompt and perception attack approaches on the vision language model, VIMA [21], which uses a multi-modal input combining both textual and visual information, allowing both prompt and perception attacks. We also perform extra evaluation over another popular robot approach embodied with the language model, Instruct2Act [20], which is included and discussed in Section 11 in the Supplementary Material due to limited space.

We perform experiments on three tasks in the VIMA-Bench environment: (1) *Visual Manipulation*, (2) *Scene Understanding*, and (3) *Rearrange*. While *Scene Understanding* is more text-dependent, *Rearrange* is more visual-dependent, and *Visual Manipulation* is the balance of both. For *Visual Manipulation*, we perform experiments over three difficulty levels, (a) *Placement Generalization*, (b) *Combinatorial Generalization*, and (c) *Novel Object generalization*, depending on the generalization level of objects and their properties based on the common-sensing abilities of the language model. Our experimental results, as detailed in Table 1, provide insightful observations regarding the impact of various attack strategies on the robot system:

**1. Different Text Attacks.** Compared to Section 5.2, results in Table 1 show extension rephrasing outperforms rephrasing attacks with more specific targets, like adjective and noun rephrasing attacks, as it lowers accuracy to 73.9%. In contrast, adjective and noun rephrasings achieve 79.9% and 76.8% accuracy reductions, respectively. Simple rephrasing less effectively drops accuracy to 83.4% and stealth rephrasing decreases the accuracy to 79.8%. This may be due to extension rephrasing introducing duplicative, confusing information that disrupts model decision-making, while the rephrasing attacks target nucleus components like nouns is more effective than others.

**2. Attacks under Different Tasks.** Table 1 illustrates VIMA's performance across three tasks under various attacks. In the *Visual Manipulation* task, accuracy falls by 15.5% and 40.1% under prompt and perception attacks, respectively. *Scene Understanding* sees minimal impact from prompt attacks (1.3% drop) but a significant 40.4% decrease under perception attacks. In *Rearrange*, VIMA faces substantial declines of 44.1% and 45.3% under prompt and perception attacks, indicating differential sensitivity to the nature of information and prompt structures across tasks.

**3. Attacks to Models with Different Robustness.** Image quality attacks have a minimal impact on the VIMA approach because VIMA is reliant to predetermined segmentation results for object detection. However, in contrast, in Instruct2Act results given in Section 11, presented in the Supplementary Material, image quality attacks substantially degraded performance from 47.4% to 12.1% in *Visual Manipulation* task. This suggests that compromising the object segmentation process in manipulation tasks can critically

| Method | Category | Attack | Placement Generalization | | | Combinatorial Generalization | Novel Object Generalization |
| | | | Visual Manipulation | Scene Understanding | Rearrange | Visual Manipulation | Visual Manipulation |
|---|---|---|---|---|---|---|---|
| Prompt | Rephrasing | Simple | 88.0 | 99.3 | 65.3 | 85.3 | 79.3 |
| | | Stealth | 86.7 | 100.0 | 55.3 | 85.3 | 70.7 |
| | | Extension | 82.0 | 98.7 | 30.7 | 81.3 | 76.7 |
| | | Adjective | 83.3 | 98.7 | 70.7 | 81.3 | 65.3 |
| | | Noun | 82.7 | 96.7 | 57.3 | 82.7 | 64.7 |
| | Average | | 84.5 | 98.7 | 55.9 | 83.2 | 71.3 |
| Perception | Image Quality | Blurring | 100.0 | 100.0 | 99.3 | 100.0 | 99.3 |
| | | Noising | 100.0 | 100.0 | 98.7 | 100.0 | 99.3 |
| | | Filtering | 100.0 | 100.0 | 98.7 | 100.0 | 99.3 |
| | Transformation | Translation | 81.3 | 80.0 | 66.7 | 82.0 | 82.7 |
| | | Rotation | 2.0 | 0.7 | 4.7 | 0.7 | 1.3 |
| | | Cropping | 5.3 | 2.0 | 6.7 | 4.0 | 0.7 |
| | | Distortion | 0.0 | 0.7 | 3.3 | 0.0 | 1.3 |
| | Object Addition | in Seg | 50.7 | 53.3 | 15.3 | 52.7 | 59.3 |
| | | in RGB | 100.0 | 100.0 | 99.3 | 100.0 | 99.3 |
| | Average | | 59.9 | 59.6 | 54.7 | 59.9 | 60.3 |
| Original | No Attack | | 100.0 | 100.0 | 99.3 | 100.0 | 99.3 |

Table 1. **Attack Results of VIMA [21] over VIMA-Bench.** We perform attack experiments over 3 tasks *Visual Manipulation*, *Scene Understanding* and *Rearrange*, while *Visual Manipulation* has been made under 3 difficulty levels: *Placement Generalization*, *Combinatorial Generalization* and *Novel Object Generalization*. **Conclusion.** VIMA framework is more vulnerable under all prompt attacks (except in the *Scene Understanding* task), and some perception attacks like transformation attacks, and the object addition attack in the segmentation image.

| Prompt \ Perception | Noising | Translation | OA in Seg | N/A |
|---|---|---|---|---|
| Simple | 88.7 | 69.3 | 46.0 | 88.0 |
| Stealth | 92.7 | 66.0 | 36.0 | 86.7 |
| Extension | 87.3 | 68.0 | 41.3 | 82.0 |
| Adjective | 90.0 | 70.7 | 50.7 | 83.3 |
| Noun | 86.7 | 62.0 | 48.7 | 82.7 |
| N/A | 100.0 | 81.3 | 50.7 | 100.0 |

Table 2. **Attack Results of VIMA [21] over different combinations of prompt and perception attacks over VIMA-Bench.** Results over all combinations of 5 prompt attacks: *Simple*, *Stealth*, *Extension*, *Adjective* and *Noun* and 3 perception attacks: *Noising*, *Translation* and *Object Addition in Segmentation*. **Conclusion.** The VIMA framework is more vulnerable under the combination of two or more different attacks.

undermine the robot system's functionality.

**4. Transformation Attacks.** A particularly noteworthy finding is the profound effect of transformation attacks, where rotation, cropping, and distortion contribute to the minimum accuracies in Table 1. Even minimal deviations, like under 10 degrees rotation or about 10 pixels shift in the perceived images, result in a complete breakdown of the language models integrated with the robotic system. These types of deviations are common in real-world settings, stemming from installation errors or manufacturing processes.

**5. Object Addition Attacks.** Furthermore, our analysis reveals that VIMA is distinctly susceptible to object addition attacks, especially addition in segmentation has an average accuracy of 46.3%. The model's heavy reliance on accurate ground-truth object segmentation for decision-making makes it vulnerable to introducing fictitious objects, which can disrupt its logical reasoning. Conversely, introducing anomalies in RGB images poses a more significant threat in systems that manually perform object segmentation.

**6. Generalization Abilities.** Table 1 analyzes *Visual Manipulation* task performance across three levels: *Placement Generalization*, *Combinatorial Generalization*, and *Novel Object Generalization*, focusing on object and texture challenges. VIMA's accuracy drops by 15.5% for *Placement Generalization* and 28.7% for *Novel Object Generalization* under prompt attacks. However, under perception attacks, the performance decrease is consistent across all levels, with about 40% drops, highlighting differential sensitivities to attack types based on generalization complexity.

**7. Consistency between Text and Perception Inputs.** Table 2 reveals that mixed attacks generally cause a greater decrease

in performance, with perception and prompt attacks together lowering accuracy by around 16% more than prompt attacks alone. Specifically, incorporating stealth rephrasing with perception attacks leads to a 21.8% fall in performance. Adding prompt attacks to noising attacks significantly drops accuracy from 100.0% to 89.1%. A similar trend is observed with translation attacks, where accuracy decreases from 81.3% to 67.2%. However, combining prompt attacks with object addition in segmentation attacks does not greatly enhance effectiveness, as it shows 6.2% additional drop in accuracy compared to using object addition alone.

For a breakdown of these experimental details, including findings and the methodologies employed, please refer to Section 8, 11, and 12 in the Supplementary Material.

### 5.4. Discussions and Take Away Messeage

From our experimental results and analysis, we derive several insights into prompt and perception attacks targeting language models integrated within robotic systems.

**1. General and target-oriented prompt attacks.** Target-oriented attacks, like adjective and noun rephrasing attacks, and stealth rephrasing attacks targeting the prompt structures, are more effective than general prompt rephrasing attacks, according to Section 5.2, #1 from Section 5.3 and Table 1.

**2. Attacks on different modalities.** Language models adjust their response based on the specific characteristics of manipulation tasks, leading to varied outcomes across different attack approaches. Specifically, prompt attacks yield more pronounced effects on tasks heavily reliant on prompts, whereas perception attacks are more impactful on tasks dependent on visual cues. This variation is evident in the results presented in Table 1 and 2, with discussion in Section 5.3, particularly in observations #2, #6 and #7.

**3. Downstream effect by attacks on perceived RGB images on object segmentation.** The attacks on perceived RGB images could lead to the failure of the object segmentation results, adversely affecting downstream perception and scene understanding tasks, as shown in Table 1 and mentioned in #3 and #5 from Section 5.3.

**4. Attacks leading to perception deviation cause significant performance drops.** Attacks causing deviations in perceived object positions can significantly reduce the task execution accuracy of robotic systems. This is true even for minor deviations caused by rotation, position, or projective errors, which are common issues in the installation of perception sensors in robotic systems, as highlighted in observation #4 from Section 5.3.

## 6. Conclusions and Open Questions

In this work, we seek to enhance the safe and effective integration of advanced language models and robotics. By conducting thorough experiments, we highlight the risks and vulnerabilities of the current state-of-the-art visual language models for robotics under adversarial attacks. We provide empirical evidence of vulnerabilities by considering several attack approaches on those models. Our findings emphasize the need for further research to ensure the secure deployment of such technologies and underscore their critical role in maintaining the safety and reliability of robotic applications.

Based on our insights and findings in this work, we list some important open problems and questions that need the immediate attention of the research community for the safe, robust, and reliable deployment of language models in robotics.

**1. Evaluation benchmarks to test the robustness of language models in robotics.** There is a need to introduce more adversarial training samples or benchmark datasets to test the robustness of the language models in robotics.

**2. Designing safeguard mechanisms.** We need a mechanism that allows robots to ask for external help under uncertainty like the mechanism proposed in [40].

**3. Explainability or interpretability of the LLM/VLM-based robotics systems**. One of the major reasons for the vulnerabilities of LLM Robotics systems against these attacks lies in the inherent black-box or/and uninterpretable components in the system (*i.e.* ChatGPT). Therefore, it is essential to identify the most vulnerable component of the pipeline to these attacks and to understand the specific vulnerabilities.

**4. Detection of Attack and Human Feedback.** A fundamental aspect of a robust and reliable system is its ability to detect attacks or vulnerabilities and subsequently signal for assistance. Therefore, developing detection strategies for LLM/VLM-based robotics systems that can identify attacks using verifiable metrics and trigger alerts for human or expert intervention becomes critical.

**5. VLM-based robotics systems with multi-modal inputs and their vulnerability.** As robot systems increasingly incorporate multi-modal inputs and large generative models, it becomes crucial to assess the vulnerabilities associated with individual modalities, such as vision, language, and audio. Equally important is identifying which components are most susceptible to attacks and under what scenarios.

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
