# OpenReview forum: "On the Safety Concerns of Deploying LLMs/VLMs in Robotics: Highlighting the Risks and Vulnerabilities"
_thecvf.com/CVPR/2024/Workshop/VLADR — VLADR 2024 Poster_

### Official Review · Reviewer_N9jz · 2024-04-21

**Rating:** 6
**Confidence:** 4

**Review:**

The manuscript addresses a critical and timely issue in the deployment of language and vision-language models within robotics.
As the integration of LLMs and VLMs in robotics becomes more prevalent, this work highlights the significant safety risks due to their vulnerability to adversarial attacks. These vulnerabilities can lead to substantial performance deterioration and pose serious safety hazards, underscoring the need for robust countermeasures to ensure safe deployment.

Pros:
1. This work conducted detailed experiments to test the vulnerability of robotics systems integrated with LLMs and VLMs. The use of multiple type of adversarial attacks across different robotics frameworks provides a broad perspective of this issue.

Cons:
1. The utilized perception attack methods are all traditional techniques such as image blurring, rotation, and object addition. This manuscript could improve by investigating some state-of-the-art attacking techniques.
2. The authors are encouraged to discuss the limitations of this work.

---

### Decision · Program_Chairs · 2024-04-22

Accept (Poster)